# Structural and Functional Insights into GID/CTLH E3 Ligase Complexes

**DOI:** 10.3390/ijms23115863

**Published:** 2022-05-24

**Authors:** Matthew E. R. Maitland, Gilles A. Lajoie, Gary S. Shaw, Caroline Schild-Poulter

**Affiliations:** 1Robarts Research Institute, Schulich School of Medicine & Dentistry, Western University, London, ON N6A 5B7, Canada; mmaitla2@uwo.ca; 2Department of Biochemistry, Schulich School of Medicine & Dentistry, Western University, London, ON N6G 2V4, Canada; glajoie@uwo.ca (G.A.L.); gshaw1@uwo.ca (G.S.S.)

**Keywords:** CTLH complex, GID complex, E3 ligase, ubiquitination, RanBP9/RanBPM, GID4, RMND5A, WDR26, muskelin

## Abstract

Multi-subunit E3 ligases facilitate ubiquitin transfer by coordinating various substrate receptor subunits with a single catalytic center. Small molecules inducing targeted protein degradation have exploited such complexes, proving successful as therapeutics against previously undruggable targets. The C-terminal to LisH (CTLH) complex, also called the glucose-induced degradation deficient (GID) complex, is a multi-subunit E3 ligase complex highly conserved from *Saccharomyces cerevisiae* to humans, with roles in fundamental pathways controlling homeostasis and development in several species. However, we are only beginning to understand its mechanistic basis. Here, we review the literature of the CTLH complex from all organisms and place previous findings on individual subunits into context with recent breakthroughs on its structure and function.

## 1. Introduction

Ubiquitination regulates proteome dynamics with exquisite specificity [1,2]. This is largely achieved by E3 ligases, which function to recruit substrates for ubiquitin (Ub) modification usually by recognizing short linear motifs (called degrons) in the target [3,4,5]. Over 600 E3 ligases exist that include either a RING (Really Interesting New Gene) or HECT (Homologous to the E6AP Carboxyl Terminus) domain that facilitates Ub transfer from an E2 conjugating enzyme to a substrate. In addition, RING and HECT E3 ligases can possess accessory domains, or form complexes that include other proteins, used to recruit substrates targeted for ubiquitination. Some multi-subunit E3 ligase complexes, such as the Cullin RING ligase (CRL) family, can use several receptor proteins that control substrate diversity and offer multiple points of regulation [6,7]. In a therapeutic context, CRL E3 ligases have been exploited as vessels for small molecules inducing targeted protein degradation; however, an area of need is expanding the toolbox to other E3 ligases, which will increase the opportunities to use these promising drugs in more clinical settings [8].

The C-terminal to LisH (CTLH) complex (in yeast, named the glucose-induced degradation deficient (GID) complex), has entered the ubiquitination field spotlight as a large and conserved multi-subunit RING E3 ligase with unique structure and functions. In *Saccharomyces cerevisiae* (*S. cerevisiae*), the GID complex is the recognin of N-terminal proline degrons (Pro/N-degron) that directs the signal-dependent ubiquitination of gluconeogenic enzymes [9,10]. In other species, we are only just beginning to understand CTLH complex mechanisms and functions. In a short amount of time, evidence of the E3 ligase activity of the mammalian complex emerged, and soon thereafter the CTLH complex was implicated in a variety of essential processes in development and homeostasis—a feature common to ubiquitin signaling. Here, we review past studies on the CTLH complex from yeast to humans and provide perspectives on the recent insights into its structure and function.

## 2. The Basics: GID/CTLH Complex Composition, Characteristics, and Conservation

### 2.1. Composition and Conservation

The CTLH complex, conserved from yeast to humans (Figure 1) [11], is characterized by its RING heterodimer, multiple protein interaction domains, and LisH, CTLH, and CRA motifs present on most protein subunits. We highlight the *S. cerevisiae* and human complexes because their structure and activity are best understood. In the *S. cerevisiae* complex (named the GID complex), Gid1, Gid5, and Gid8 function together as the scaffold to support the organization of other protein subunits. This scaffold, the RING heterodimer comprising Gid2 and Gid9, and an interchangeable substrate receptor (Gid4, Gid10, or Gid11) form the minimal stable GID complex, termed GID^SR4^, GID^SR10^, or GID^SR11^ [12,13,14]. This complex can recruit Gid7, which has the ability to stimulate oligomeric complex formation [15]. In humans, the CTLH complex is named RanBP9 (also known as RanBPM; homologue of Gid1), GID8 (also known as TWA1; homologue of Gid8), and ARMC8 (similar to Gid5) act as the scaffold; RMND5A (homologue of Gid2) and MAEA (homologue of Gid9) are the RING heterodimers required for E3 ligase activity; and GID4 (homologue of Gid4) is a presumed substrate receptor. WDR26 (homologue of Gid7) and/or muskelin bind RanBP9 and facilitate oligomerization of the complex [15]. Additionally, paralogues of RMND5A and RanBP9, RMND5B and RanBP10 have been implied as human complex members [15,16]. Human YPEL5 and its orthologues (Moh1 in yeast) are also part of the complex, but their role is unclear [15,16,17,18,19]. Key to the activity of the complex, Gid2/RMND5A and Gid9/MAEA provide a unique RING domain heterodimer that can bind the E2 enzyme UBE2H (Ubc8 in *S. cerevisiae)* and stimulate E2-catalyzed Ub transfer to a recruited substrate [11,12,15].

Several important differences make the human complex distinct from the yeast version. Firstly, muskelin is not encoded in the yeast genome and there is only one gene, instead of the two paralogues, for RanBP9/10 and RMND5A/B [11]. Second, in contrast to the *S. cerevisiae* complex, interchangeable substrate receptors with GID4 have yet to be identified in other eukaryotes, although evidence of substrate engagement independent of GID4 has emerged [20]. Finally, although yeast Gid5 is not a true homologue of ARMC8 [21], structurally they are similar [15]. An important difference, however, is that two ARMC8 isoforms (α and β), instead of one Gid5 in yeast, associate with the human complex [10,22,23], but only the α isoform can bind GID4 [20]. The presence of these differing components in other species likely gives the CTLH complex distinct functionalities as compared with the *S. cerevisiae* GID complex.

### 2.2. Domains in the GID/CTLH Complex: Structure and Roles in Complex Activity, Formation, or Substrate Engagement

The distinguishing characteristic of GID/CTLH complexes, and the origin of the CTLH name, is the intertwining conserved LisH, CTLH, and CRA motifs. These regions comprise between two (LisH) and four (CRA) α-helices, as noted previously in three-dimensional structures of the splicing protein SMU1, the transcriptional co-repressor proteins TOPLESS and TOPLESS-related protein 2, and the gene product of *LIS1*, which is mutated in classical lissencephaly (Figure 2a–d) [24,25,26,27]. In these proteins, LisH forms a two-helix hairpin that mates with LisH from an adjacent protein in an antiparallel arrangement to promote oligomerization (Figure 2a). C-terminal to the LisH, three CTLH α-helices connect it to the CRA motif which has a more extended conformation (Figure 2b,c). In the TOPLESS proteins, the two N-terminal helices of adjacent CRA motifs dimerize with each other at roughly 90°. As observed in the structures of SMU1 and TOPLESS dimers, the last CRA helix of each monomer cross over, forming an X shape adjacent to the LisH hairpin (Figure 2d).

In the GID/CTLH complexes, Gid1 (RanBP9/10), Gid2 (RMND5A/B), Gid7 (WDR26), Gid8 (GID8/TWA1), Gid9 (MAEA), and muskelin contain LisH, CTLH, and CRA motifs (Figure 2e). The smallest CTLH subunit, GID8, only contains these structures; it serves as an essential core complex member in the scaffold where the helices are used to bind multiple subunits (Figure 2f) [12,15]. The LisH helices from Gid1 and Gid8 are essential to pair these two proteins. The arrangement and orientation of the LisH–CTLH–CRA motif in Gid1 is similar to those found in Smu1 and TOPLESS, whereas the orientation of this triad appears to be altered in Gid8.

Several other protein–protein interaction domains are present on CTLH complex subunits. The main difference between human RanBP9 and RanBP10 is that RanBP9 has poly-glutamine and poly-proline sequences at the N-terminus, but RanBP10 does not, a difference with functional consequences on regulating MET receptor signaling (Figure 2e) [28]. Crystallography shows that the RanBP9 and RanBP10 SPRY domain (also present on Gid1) are nearly identical, with two antiparallel β-sheets held together by hydrophobic and polar interactions, a helix present at each terminus, and a shallow binding pocket (Figure 2f) [15,29]. For RanBP9, the SPRY domain mediates interaction with most of RanBP9′s [15,29]. For RanBP9, the SPRY domain mediates interaction with most of RanBP9′s many associated proteins [30].

Gid5/ARMC8 does not have LisH, CTLH, or CRA motifs. Instead, several helical armadillo (ARM) repeats are present. In general, these repeats fold together to form a superhelix of helices that serves as a versatile protein interaction surface (Figure 2f) [31]. In the GID/CTLH complex, one-half of the Gid5/ARMC8 ARM repeats engages Gid1/RanBP9–GID8/TWA1 in the scaffold, whereas the other half anchors GID4 [12,15,20].

RING domains, present on Gid2/RMND5A and Gid9/MAEA, typically bind an E2 enzyme and promote the E2~Ub transfer. The canonical RING structure is a cross-braced arrangement, with cysteines and a histidine coordinating two zinc ions critical for its compact α/β fold [32]. Details of the GID/CTLH RING structures have been best described by cryo-EM structures of the *S. cerevisiae* complex, where Gid2 (RMND5A homologue) adopts a unique heart-shaped RING domain encompassing a single zinc (instead of the typical two for RING domains) and Gid9 (MAEA homologue) with a “RING-like” (RING-L) domain that does not coordinate zinc on its own (Figure 2e,g) [15]. Gid2 and Gid9 dimerize through their C-terminal RING and RING-L domains, as well as through an intertwining coiled-coil structure at their N-termini which is reminiscent of the coiled-coil found in the BRCA1/BARD1 RING heterodimer structure [33]. This overall arrangement stabilizes the Gid2–Gid9 dimer structure, which explains the in vivo interdependence of Gid2/RMND5A and Gid9/MAEA in yeast and human cells [10,22,34].

In *S. cerevisiae*, Gid4 and Gid10 antiparallel β-barrels recognize N-terminal prolines as part of the Pro/N-degron pathway (Figure 2h,i) [3,9,12,14,35,36,37]. A PGLW peptide, resembling the yeast Pro/N-degron, fits snugly at the bottom of a deep and narrow binding cleft in the human GID4 β-barrel in a precise position to mediate a network of hydrogen bonds [35]. Other hydrophobic residues can be accommodated in the binding cleft, although the downstream sequence context is critical, particularly for residues in positions 2 and 3 [37,38]. Despite the sequence diversity in GID4 degron binding preferences observed using in vitro experiments, no GID4 substrate has currently been definitively determined outside of budding yeast. Slight structural differences in *S. cerevisiae* Gid10′s β-barrel enable the binding of a bulky hydrophobic residue in position 2 of the degron (as opposed to smaller Gly/Ala preferred for GID4), such as for its only known target thus far, Art2 (Nt-Pro-Phe-Ile-Thr) [36,37,39]. Gid11, the third *S. cerevisiae* interchangeable substrate receptor, recognizes proteins with an N-terminal Thr [13]. Alphafold predicts a β-propellor-like structure present in Gid11 [40], but how this captures Nt-Thr substrates needs further study.

WD40 repeats are present on Gid7/WDR26 homologues, forming an atypical β-propeller (Figure 2f) [11,15]. A structurally similar six-blade kelch repeat is predicted for muskelin. Both WD40 and kelch β-propellers facilitate protein–protein interactions or protein–DNA interactions and are often found in multi-subunit complexes, including other E3 ligases [3,41]. Additionally, muskelin has a discoidin domain at the N-terminus before the LisH domain (Figure 2a,g) [11]. Crystal structures show that the mouse muskelin discoidin domain, which is highly conserved in mammals and shares 53% identity with its *Drosophila melanogaster* homologue, forms a jellyroll fold, comprising two antiparallel β sheets (a five- then three-stranded β-sheet) facing each other with a hydrophobic core (Figure 2g) [42,43]. In other proteins, discoidin domains exhibit a variety of protein interactors, but also a wide range of other types of interacting molecules, such as lipids, phospholipids, galactose, and collagen [44].

Most of the CTLH complex protein interaction domains described above act as substrate recruitment modules in other E3 ligase proteins [3]. At present, however, a substrate recruitment function has only been demonstrated for the yeast Gid4/Gid10 β-barrel. Interestingly, the WDR26 β-propeller has been proposed to act as a substrate receptor for the complex binding target HBP1 [20], although this awaits structural validation. Perhaps muskelin also recruits substrates, either via its kelch repeat β-propeller or its discoidin domain, which bears resemblance to the Gid4/Gid10 β-barrel. In fact, preliminary evidence exists for muskelin to bind targets in *D. melanogaster*, which does not have a Gid4 homologue [45]. The RanBP9/10 SPRY domain-binding pocket could conceivably also bind targets. If not by themselves, these domains could be contributing in a multivalent manner to substrate-binding, strengthening the interaction and/or helping orient lysines to the E2 active. Alternatively, they may facilitate internal complex interactions (as the ARMC8/Gid5 ARM repeats do), be sites for regulation, or anchor/target the complex to specific subcellular locations or organelles. Clarifying the roles of each CTLH complex subunit for targeting substrates will be essential for the development of chemical tools designed to manipulate this multi-subunit E3 ligase.

## 3. Complex Architecture and Activity

### 3.1. Architecture and Interactions

An initial 2D topology of the yeast GID complex was provided by protein interaction assays in vivo using yeast deletion strains [10,34,46]. Since then, a more complete picture has emerged with cryo-electron microscopy (cryo-EM) data revealing that the yeast GID^SR4^ complex assembles as a clamp-like structure with Gid4 at one end of the clamp, a T-shaped Gid2–Gid9 RING dimer on the other end, and a core scaffolding unit of Gid1, Gid8, and Gid5 that connect the two ends together (Figure 3a) [12]. It provided the first model for Ub transfer by the GID/CTLH complex: dimeric Mdh2 substrate, with its proline N-terminus engaged with the Gid4 β-barrel, fills a 50 Å gap between Gid4 and the catalytic module. In this arrangement, Mdh2 lysine residues to be ubiquitinated are proximal to the active site of the E2 enzyme Ubc8, which is bound to Gid2. Residues from the Gid2–Gid9 RING dimer stabilize the Ubc8~Ub conjugate, activating the transfer of Ub from the E2 to the accessible lysine residues of Mdh2.

In the scaffold of the yeast complex, Gid1–Gid8–Gid5 interactions occur through multiple surfaces: (1) Gid1 and Gid8 through each other’s LisH and CRA (C-terminal segment) domains; and (2) a surface comprising parts of Gid1′s SPRY, CTLH, and N-terminal CRA portions and parts of Gid8′s CTLH and N-terminal CRA portions are engaged by the N-terminal half of Gid5′s ARM repeats [12]. Complex stability is compromised in yeast deficient of Gid1 or Gid8, highlighting the essential role for both proteins as the core of the complex [46]. The same is true for the human homologues, RanBP9 and GID8/TWA1 [22]. Meanwhile, a concave surface formed by C-terminal ARM repeats of Gid5 is occupied by Gid4. A C-terminal sequence shared between Gid4 (yeast and human), Gid10, and Gid11 of an acidic residue sandwiched between hydrophobic residues is required for their interaction with Gid5 [12,13]. The RING subunit Gid9 appears to use the N-terminus of its CRA motif to contact the CRA of Gid8, linking the scaffolding unit with the RING proteins [12]. This small interface may give rise to suspected mobility of the Gid2/Gid9 RING segment relative to the remainder of the GID scaffolding unit.

The initial cryo-EM GID complex structural determination lacked Gid7 [12]. Remarkably, a more recent study showed that Gid7 induces oligomerization of the recombinant yeast complex to yield a 20-protein oval-shaped structure of 1.5 MDa with two opposing RING-bound E2~Ub active sites, two Gid7 homodimers, four copies of a Gid1–Gid8 dimer, and two Gid4–Gid5 dimers (Figure 3b) [15]. At opposite ends of the Gid7 dimer on two sides of the oval, Gid1 binds a Gid7 protomer heterodimerizing via both their CTLH domains and N-terminal component of their CRA domains. The LisH and C-terminal part of the CRA domain of Gid7 mediates its homodimerization. In the cavity of the oval, Gid1 SPRY and two Gid7 β-propellers are adjacent to each other on two sides, and on the other two sides of the oval, a Gid4 β-barrel is next to one of the RING heterodimers. Its design accommodates the tetrameric substrate Fbp1 in the center with two of its Pro/N-degrons bound to the two Gid4 molecules, with Fbp1’s lysines positioned in the active site of two RING-bound Ubc8 E2 enzymes. Despite having lower intrinsic activity compared with its monomeric form, ubiquitination by this oligomeric Gid7-induced complex, termed supramolecular “chelator” complex, is kinetically favoured for the tetrameric state of Fbp1 [15].

For human proteins, the monomeric complex architecture is similar to the yeast complex where a RanBP9–GID8–ARMC8 scaffold replaces Gid1–Gid8–Gid5, the RMND5A-MAEA RING module replaces the Gid2–Gid9 RING heterodimer, and human GID4 is a presumed substrate receptor (Figure 3c) [15,20]. The human complex can also oligomerize and adopt a ring or oval-like shape with WDR26, the homologue of Gid7, mediating dimerization interfaces and being required for higher-order assemblies of the CTLH complex in vivo (Figure 3d) [15,20]. Unique to the human complex, tetrameric muskelin directly binds RanBP9–GID8, similarly to WDR26, although currently it is unclear in vivo if there are WDR26- or muskelin-exclusive complexes (Figure 3e) [15]. Another unique feature is four ARMC8–GID4 modules present compared with two in yeast, thus potentially providing more N-degron–GID4 binding sites [15]; however, whereas both ARMC8 isoforms can integrate into the complex, only α can bind GID4 [20]. Thus, if there are various oligomeric complex assemblies containing different combinations of ARMC8 isoforms (e.g., a complex with all β or all α isoforms, or a mixture of them), the number of GID4 molecules present could range from 0 to 4 (Figure 3f shows an example).

Given that CTLH complex proteins sediment in multiple fractions in a sucrose gradient [15,23], it appears that there may be multiple different configurations of alternate subunit assemblies. An important step forward is to determine the details of the in vivo chelator supramolecular complex with respect to function and regulation. Many questions remain unanswered regarding how many variations of the complex exists, whether the complex substructures can be favored or induced by specific signals to ubiquitinate certain substrates, and what the possible roles are of the versatile protein–protein interactions present on the different subunits in the supramolecular structure. A recent report showed that the CTLH complex profile appears to switch from RanBP9 to RanBP10 during the course of erythropoiesis, providing the first evidence that distinct CTLH complex assemblies can be modulated [47].

Post-translational modifications (PTMs) are likely involved in regulating complex formation. Muskelin is subjected to protein kinase C (PKC)-dependent phosphorylation, which appears to inhibit its multimerization, as well as CTLH complex-dependent poly-ubiquitination causing its proteasomal degradation [22,48]. RanBP9 is phosphorylated in response to DNA damage at Ser 603 and directly by ataxia telangiectasia-mutated (ATM) in vitro at Ser 181 and 603 [49,50]. Ionizing radiation (IR) treatment also results in Ranbp9 rapid accumulation in the nucleus which was shown to be dependent on ATM kinase activity [49]. These few examples, with more expected in future studies, suggest that PTM regulation of the CTLH complex could temporally and spatially control subunits, and by extension modulate the assembly of specific subcomplexes. Many other PTMs have been identified in high-throughput studies (listed in PhosphoSitePlus [51]), but their effects on complex activity and function remain to be elucidated.

### 3.2. The RING Heterodimer: Structure and Activity

The first documentation of E3 ligase activity for the GID/CTLH complex was reported in 2008 using an in vitro ubiquitination assay with recombinant *S. cerevisiae* Gid2 [10]. Since then, the human, *Xenopus laevis*, and *Lotus japonicus* homologues of Gid2/RMND5A have all now been demonstrated to have intrinsic ligase activity [10,22,52,53]. Additionally, ligase activity has now been demonstrated for complete human and *S. cerevisiae* complexes [12,15,16,22].

Zinc binding is critical to the Gid2 RING domain: ligase activity for the *S. cerevisiae* and *X. laevis* is abolished if a cysteine involved in the zinc coordination is mutated (C379 in yeast Gid2, equivalent to C354 in human RMND5A), likely because the domain cannot fold properly [10]. Additionally, other critical residues within the Gid2/RMND5A RING domain include the hydrophobic E2-binding site (Gid2: V363, L364, human RMND5A: I338, L339) or the linchpin (Gid2: K365, human RMND5A: R340), which is known to promote Ub transfer by forming hydrogen bonds with both Ub and the E2 that restricts their relative orientations [32,54]. Although Gid9/MAEA does not bind the E2, Y514 at the extreme C-terminus of Gid9 (equivalent to Y394 of human MAEA) acts as a non-RING priming element required for activity [12,15], akin to other inactive RING partners in RING heterodimers (e.g., BARD1) [55].

Successful in vitro ubiquitination assays with the complex have either used the promiscuous UBE2D2 (ubiquitin-conjugating enzyme E2 D2, aka UbcH5b) or Ubc8 (human homologue: UBE2H, aka UbcH2) as the E2 [10,12,15,22,52,53]. Ubc8 and its human homologue contain an acidic C-terminal extension similar to yeast CDC34, and thus are class III E2 enzymes [56]. In a yeast two-hybrid screen, RMND5B (RMND5A and MAEA were not tested) showed positive interactions with UBE2D2, UBE2D3, UBE2D4, UBE2E1, UBE2E3, and UBE2W, but surprisingly not UBE2H [57]. Most relevant is E2 used by the complex in vivo for a particular substrate and which polyubiquitin linkages it generates, which must be investigated on a case-by-case basis. The data thus far for the yeast or human complex indicate that K48 polyubiquitin chains are generated with UBE2H/Ubc8 as the E2, whereas both K48 and K63 polyubiquitin chains can be generated with UBE2D2 as the E2 [12,15,22].

## 4. What Started It All: Functions of the Yeast GID Complex in Glucose Metabolism and Beyond

The yeast GID complex biochemical properties and function have been elegantly characterized by studies led by Dieter H. Wolf and Hui-Ling Chiang. Due to the evolutionary conservation of CTLH complex subunits in eukaryotes [11], it set the foundation for investigating the complex in other species.

### 4.1. Catabolite Inactivation

In glucose-starved *S. cerevisiae* cells undergoing gluconeogenesis, the replenishment of glucose induces immediate inhibition and degradation of the gluconeogenic enzyme fructose 1,6-bisphosphatase (Fbp1, also known as FBPase) [58]. It is part of a process called catabolite inactivation, whereby *S. cerevisiae* switch from gluconeogenesis back to a normal state of glycolysis.

Interestingly, Fbp1 degradation involves either the ubiquitin–proteasome system if starvation is ≤24 h, or via the vacuole (homologous to the mammalian lysosome) if starvation is 1–3 days (Figure 4) [10,59,60,61,62,63,64,65,66,67,68,69]. Later, both degradation mechanisms were linked to additional gluconeogenic enzymes, Mdh2, Icl1, and Pck1 [10,34,60,67,70]. In either case, early genetic screens from separate studies identified factors that remarkably overlapped in both Fbp1 degradation mechanisms, being named either vacuolar import and degradation (VID) genes in the long glucose starvation condition or the GID genes in the short starvation condition [60,69]. To avoid confusion, in this review we refer to the genes/proteins with the more frequently used GID designation.

### 4.2. GID-Mediated Proteasomal Degradation of Gluconeogenic Enzymes

Starving *S. cerevisiae* for 24 h followed by the re-introduction of glucose causes rapid Fbp1 ubiquitination in the cytosol, followed by 26S proteasomal degradation [59,60,61]. Investigations of this process revealed the >600 kDa GID complex to be an E3 ligase that polyubiquitinates Fbp1 and the other gluconeogenic enzymes through Gid2–Gid9 RING heterodimer binding to the E2 Ubc8 [10,12,34,65]. A key step is the rapid induction of *GID4* expression after glucose replenishment that triggers ubiquitination of the metabolic enzymes by acting as the substrate receptor [9,10,12]. This signal-dependent ubiquitin activation depends on the Gid4 molecular recognition of N-terminal proline degrons on the gluconeogenic enzyme substrates [9]. Glucose-induced degradation of Pro/N degron containing metabolic regulators Acs1 and Aro10 is also dependent on the GID complex [71]. For Pck1 and Aro10, aminopeptidases must first trim their N-termini residues to expose the degron [72]. Interestingly, once cells return to a normal metabolic state, Gid4 itself is ubiquitinated and rapidly degraded, a process which is dependent on the proteasome and GID complex proteins [10,73]. The AAA ATPase cdc48 (homologue of VCP in mammals), is a cofactor complex of Ufd1–Npl4, and the ubiquitin receptors Dsk2 and Rad23 are also required for Fbp1 degradation, likely functioning to facilitate the delivery of polyubiquitinated Fbp1 to the proteasome [74].

### 4.3. GID-Mediated Vacuole Degradation of Gluconeogenic Enzymes

When cells are instead starved in ethanol for longer than 24 h, Fbp1 is secreted as extracellular vesicles in the periplasm [75,76]. Upon glucose replenishment, Fbp1 is internalized and becomes localized to actin patches, which is dependent on the endocytosis proteins End3, Sla1, Arc18, and the PI3 kinase Vps34 [75,77]. At actin patches, 30–50 nm ‘Vid’ vesicles are formed with Fbp1 in the lumen and Gid1, Gid4, Gid5, and coatomer COPI proteins such as Sec28 on the periphery [68,78,79,80]. The vesicles then dissociate from actin and subsequently cluster together to deliver Fbp1 to the vacuole for degradation [67,68,70,79,81]. As in the short starvation mechanism, *GID4* is induced after glucose replenishment, and blocking its translation with cycloheximide prevents Fbp1 vacuolar degradation [64,78]. As in the proteasome pathway, the N-terminal proline of the gluconeogenic enzymes is required for their vacuolar degradation [67]. Gid2 and Gid8, K48/K63 polyUb chain formation, and the E2 enzyme Ubc1 are also required for Fbp1 vacuolar degradation [67,82]; however, the ubiquitin ligase activity of the GID complex in this context has not been addressed.

A few factors are unique to each mechanism. The secretion of Fbp1 to the periplasm during long starvation is likely the cause of Fbp1 being delivered to the vacuole, instead of the proteasome. Additionally, cAMP signaling is required for Fbp1 phosphorylation and the fusion of Vid vesicles with the vacuole, but is not required for Fbp1 proteasomal degradation [67]. In a rare case of their non-redundant functions, two members of the Hsp70 family of chaperones are differentially required for either degradation mechanism: Ssa1 interacts with Fbp1 and is required for the 24 h glucose-induced ubiquitination and proteasomal degradation, whereas Ssa2 is required for the import of Fbp1 into the Vid vesicles [83,84,85]. The specificity for Ssa2 in the Fbp1 Vid vesicle–vacuolar degradation pathway has been mapped to G83 in its nucleotide binding domain, one of the few residues that is different (albeit only by a methyl group) between Ssa1 and Ssa2 (A83 in Ssa1) [86].

### 4.4. Functions and Targets beyond Catabolite Inactivation

The GID complex role in catabolite inactivation seems to be limited to yeast, and further limited to *S. cerevisiae* as the Pro/N-degrons in Fbp1, Mdh2, and Icl1 are masked or altered in other species [87]. The pathogenic yeast *Candida albicans*, for example, does not undergo catabolite inactivation [88,89]. Consequently, *C. albicans* species have metabolic flexibility, which promotes resistance to macrophage killing, host colonization, and virulence [89]. This implies that the evolution of *C. albicans* prioritized infection ability over ATP efficiency. The deletion of GID genes in *S. cerevisiae* impairs its catabolite inactivation and renders those cells with metabolic flexibility and virulence, similar to *C. albicans* [89].

Other functions in addition to glucose metabolism have been described for the GID complex in *S. cerevisiae*. Cells with deletions of any Gid protein except Gid7 are hypersensitive to rapamycin treatment, implying a role for the non-chelator complex in mTOR signaling [90,91]. Indeed, at least Gid1, Gid2, and Gid5 are required for efficient rapamycin or nitrogen starvation-induced internalization and degradation of plasma membrane-bound hexose transporter Hxt7 [90].

Unlike Gid4, Gid10 is not expressed under normal conditions or during glucose recovery, but is induced by heat shock, osmotic stress, or starvation of nitrogen or individual amino acids [12,14,39]. During heat shock, the N-terminal proline containing Art2 is a target of GID^SR10^, its first one identified [39]. The regulation of Art2 by GID^SR10^ during heat shock in part affects the Rsp5-dependent import and degradation of amino acid transporters Lyp1 and Can1. The GID-dependent degradation of the previously mentioned Hxt7 transporter also requires Rsp5, as does the GID-dependent, glucose-starvation-induced import and degradation of Hxt3 [90,92]. Thus, GID^SR10^ may function through a common mechanism (targeting Art2) to regulate plasma membrane receptors but in different stress responses.

Gid10 induction by various stressors is only temporary: Gid10, similarly to Gid4 during glucose recovery, is a target of the GID complex itself and is quickly degraded [39,73]. The purpose of the negative autoregulation is seemingly to have the complex available to whichever substrate receptor is induced by distinct environmental perturbations, thus poised to quickly maintain homeostasis.

## 5. Functions and Ubiquitination Targets of the CTLH Complex from Drosophila to Humans

Since the establishment of the complex as an E3 ligase, discoveries of putative or in vitro confirmed ubiquitination targets of the mammalian complex have come to light, such as transcription factor HBP1, nuclear matrix protein Lamin B2, energy regulator AMPK, glycolysis enzymes PKM2 and LDHA, and its own subunit muskelin [16,22,93,94,95]. These findings have implicated the GID/CTLH complex in a number of critical functions in different organisms, such as zygote development in *D. melanogaster*, nodule organogenesis in *Lotus japonicus* plants, organismal lifespan in *Caenorhabditis elegans*, neurodevelopment in *X. laevis*, and erythrocyte differentiation in mammals (Figure 5) [45,47,52,53,93,94,96,97,98]. Unlike in yeast, however, in-depth mechanisms of substrate capturing and ubiquitin transfer have not been realized thus far.

Many studies have described interacting proteins and the effects of overexpression and/or knockdowns for various complex subunits in different model systems, most prominently for RanBP9/RanBPM (reviewed in [30]); however, the stoichiometric relationship between subunits must be considered. There is an interdependence in the protein levels of core CTLH complex subunits that was revealed upon the downregulation or knockout of individual subunits [16,22]. What is unclear is whether the stoichiometry of the complex adapts to the overexpression of individual subunits or if it may promote specific complex assemblies. Furthermore, RNA to protein correlation of CTLH complex subunits across cancer cell lines is very low, so caution must be applied for interpretations of altered expression of a subunit if only RNA levels are considered (Table 1) [99].

Nevertheless, previous work focused on individual subunits should be re-assessed in light of the current realization that these proteins are part of a multi-subunit E3 ligase complex. Most complex subunits including the E2 UBE2H do rank in the top genetic co-dependencies of each other in the Cancer Dependency Map project [100], confirming the common functions of subunits. Here, we summarize the past work on functions of the CTLH complex subunits into unifying themes, with emphasis on the recent findings of targets.

### 5.1. Differentiation and Development

In multiple animal models, various complex subunits have been ascribed functions in developmental and cell differentiation pathways. In mice, RanBP9 knockout resulted in both sexes being sterile due to defects in oogenesis and spermatogenesis [101,102]. In *D. melanogaster*, two groups exhibited fascinating function and regulation of the entire complex as part of the precise temporal control of the maternal proteome in the maternal-to-zygotic transition (MZT). In the early stages of the MZT, the *D. melanogaster* CTLH complex is activated by translational upregulation of the UBE2H homologue, causing the CTLH-dependent degradation of RNA-binding components of a translation-inhibiting complex required for oogenesis [45,96].

WDR26, RanBP9, and RMND5A have all been individually linked to brain development in a variety of model organisms. In *X. laevis*, a species in which Rmnd5 E3 ligase activity has been demonstrated, both Rmnd5 and wdr26 are expressed early and throughout embryonic development, and both show highest expression in the neural regions [52,97]. Deficiency of both Rmnd5 and wdr26 caused a forebrain formation impairment and reduction in the same neural marker, *pax6* [52,97]. In zebrafish embryos, RanBP9 expression is also highest in the neural regions and its deficiency caused defects in brain development and retinogenesis [103]. Strikingly, several reports showed that global *RanBP9^−/−^* mice have neonatal lethality and postnatal growth inhibition, due at least in part to a compromised somatosensory system [101,102,104]. In humans, WDR26 frameshift and nonsense or missense mutations in LisH, CTLH, and WD40 domains are observed in Skraban–Deardorff syndrome, a unique, newly discovered neurodevelopmental disorder associated with intellectual impairment, seizures, and distinctive facial features [105]. Taking all these phenotypes together, it is likely that the complex has a critical role in an early stage of neurodevelopment; for example, it could be mediating a ubiquitination event required for proper neural differentiation.

An important role for complex members in red blood cell homeostasis has been well documented. An initial study showed that *Maea^−/−^* mouse embryos died perinatally with anemia and differentiation defects in erythroid and macrophage lineages, primarily caused by defective erythroblast enucleation [106]. A recent study, however, observed no perinatal lethality, anemia, or enucleation defect in young adult *Maea^Csf1r-Cre^* mice [98]. Instead, macrophage development, erythroblastic islands formation, and erythroblast maturation was impaired if *Maea* was deleted specifically in the monocyte–macrophage lineage, but this phenotype was not observed if *Maea* was deleted in the erythroid lineage. This suggested that Maea instead is critical in macrophages only. WDR26 has also been associated with regulating red blood cell development. *Wdr26* expression is upregulated in terminally differentiating erythroblasts and its knockdown caused severe defects in enucleation, a reduction in hemoglobin production, and blocked differentiation at the basophilic erythroblast stage [93]. Furthermore, *Wdr26^−/−^* zebrafish exhibited profound anemia likely due to defective erythropoiesis, a phenotype also reported in the initial study on *Maea^−/−^* mouse embryos [93,106]; however, rather than observing defects in erythroblastic island adhesion or macrophage differentiation, the Wdr26 knockout animals had deficiencies in the nuclear opening of erythroblasts. This led to the discovery that the CTLH complex directs the polyubiquitination and degradation of lamin B, which facilitates enucleation [93].

RanBP10 has also been linked to blood cell homeostasis. *RanBP10^−/−^* mice are viable and have no obvious phenotype, but do have defective hemostasis, platelet activation and aggregation, and impaired thrombus formation [107,108]. Slight decreases in erythrocyte numbers and size were observed [107]—an anemic-like phenotype shared by *Wdr26^−/−^* zebrafish and the first report on *Maea^−/−^* mice [93,106]. Finally, *GID4* was identified as a novel gene required for hematopoietic stem/progenitor cell specification, but this has yet to be investigated in detail [109].

Overall, there is a clear importance of CTLH complex subunits in different aspects of development and differentiation. Thus far, however, the ubiquitination activity has only been linked to the degradation of RNA binding proteins during MZT in *D. melanogaster* and degradation of lamin B for nuclear condensation in differentiating mammalian erythroblasts [45,93,96]. More mechanisms and ubiquitin targets in developmental contexts are likely to be revealed soon.

### 5.2. Cell Migration and Adhesion

The mammalian complex has been associated with several cell migration and adhesion pathways. Reports have shown RanBP9 association with various integrin, junctional, receptor, and adhesion proteins (reviewed in [30]). The depletion of RanBP9 increased HT22 and NIH3T3 cell attachment by disrupting focal adhesion signaling [110] and breast cancer cell invasiveness by regulating BLT2-mediated reactive oxygen species generation and IL-8 production [111]. Muskelin was initially identified in a screen for proteins that promoted C2C12 mouse myoblast cell line adherence to a thrombospondin-1 substratum [112]. In rat lens epithelial cells, muskelin depletion reduced Rho-GTP activation, myosin phosphorylation, the dissociation of stress fibers, and cell migration [113]. Muskelin and RanBP9 depletion in lung A549 cells adherent on fibronectin caused enlarged cell perimeters and altered morphology and F-actin distribution [114]. WDR26 has been linked with cell migration in multiple cell types, but with opposing effects observed. In leukocytes, WDR26 is required for SDF1α-induced cell migration and promotes PI3K/Akt-signaling-mediated migration and invasiveness in MDA-MB-231 breast cancer cells [115]. In intestinal epithelial cells, however, WDR26 was found to inhibit FPR1-mediated cell migration and wound healing [116].

Thus far, the only direct implication involving the entire complex in cell migration is through a negative regulation of histone deacetylase 6 (HDAC6) activity, which is likely responsible for the increased cell migration observed in RanBP9-deficient HEK293 cells [117]. Cells depleted of RanBP9, muskelin, and RMND5A showed increased HDAC6 activity and/or increased deacetylation of HDAC6 target α-tubulin, but no change in HDAC6 protein levels, whereas RanBP9, MAEA, and GID8/TWA1 were shown to be colocalized at microtubules with HDAC6 [117]; however, in this context, ubiquitination was not investigated, so the regulatory mechanism of HDAC6 by the CTLH complex remains unclear. The ubiquitination of HDAC6 that alters its activity or the ubiquitination of an HDAC6 coregulator are two possible mechanisms underlying HDAC6 regulation by the CTLH complex.

### 5.3. Nuclear Functions

As already mentioned, the CTLH complex is implicated in the nuclear condensation of developing erythroblasts via direct polyubiquitination of lamin B [93]. Beyond this, an exact nuclear role is unclear, but chromatin regulation is likely because at least two complex members have been found together in the interactomes of several critical transcription factors or DNA repair proteins [118,119,120,121,122,123]. Furthermore, UBE2H has been linked to histone ubiquitination [56,124,125], but whether this involves the CTLH complex is unknown.

In support of a role in transcription, microarray analyses of RanBP9 Hela and HCT116 knockdown cells indicated numerous effects on gene expression [126]. RanBP9 and/or RanBP10 interactions with steroid and hormone nuclear receptors, such as the androgen receptor and glucocorticoid receptor, have been observed, and both have been shown to act as transcriptional co-activators for these proteins [127,128,129]. RanBP9 also interacted and enhanced transcriptional activities of Epstein–Barr virus (EBV) proteins Rta and Zta, and was present on Zta-responsive elements on EBV gene promoters [130,131]. Sumoylation of the viral transcription factors by Ubc9 was regulated by RanBP9, which affected their transcriptional activity [130,131]. Thus far, that is the only established direct mechanism for any complex member on transcriptional regulation.

### 5.4. Cell Proliferation, Death, and Survival Pathways

Pro- and anti-proliferative functions have been documented for either the entire complex or subunits individually, particularly through regulation of the MAPK and WNT pathways. Lampert et al., 2018, described decreased cell proliferation in newly generated WDR26 and MAEA knockout retinal pigment epithelium cells manifested by the downregulation of cell cycle markers, which then adapted to be indistinguishable from control cells after several days of culturing [16]. This effect on cell proliferation was attributed to a CTLH-complex-dependent regulation of the protein stability of HBP1, a transcription factor that regulates the expression of cell cycle regulators. In an in vitro ubiquitination assay with UBE2H as the E2, HBP1 was ubiquitinated by the recombinant CTLH complex. This confirmed that HBP1 is a direct ubiquitination target of the complex, the first non-yeast substrate identified. The WD40 repeats of WDR26 are required for the binding and ubiquitination of HBP1, but GID4 is not, providing important evidence that the human complex can engage substrates independent of GID4 [20]. Increased cell growth in contrast to what was reported in Lampert et al., 2018 [16], was observed in RMND5A knockout HEK293 cells and RanBP9-depleted HEK293 cells and mouse embryonic fibroblasts [132,133]. Furthermore, the downregulation of RanBP9 promoted tumour formation in a mouse xenograft model [133]. In these contexts, the regulation of c-Raf kinase protein levels and the downstream activation of MEK1/2 and ERK1/2 phosphorylation were suggested to have a contributory role to the phenotype [132]. c-Raf was shown to undergo RMND5A-dependent ubiquitination, but whether this involves direct ubiquitination by the CTLH complex or another E3 ligase is not known [133].

Despite the overall conservation between the yeast and mammalian complexes, the mammalian (human or mouse) complex does not regulate gluconeogenesis, and does not ubiquitinate human Fbp1, likely because, as already mentioned, the degrons are not the same [16,134,135]. Instead, the human complex has been demonstrated to inhibit the opposite pathway, glycolysis, by regulating the ubiquitination of enzymes PKM and LDHA [95]. Instead of degradation, however, PKM and LDHA activities were increased in RanBP9-deficient cells, and global proteomic and ubiquitinome analyses suggested that non-degradative ubiquitination by the complex may be prevalent [95]. A corresponding increased glycolytic flux and altered metabolism was observed in RanBP9-deficient HeLa cells [95], a hallmark of cancer cells which enables them to survive as highly proliferating cells [136].

Increased autophagic flux linked to reduced mTOR activity was observed in RMND5A knockout NIH-3T3 cells [94]. This regulation was reported to occur through RMND5A-dependent K48 polyubiquitination and the degradation of AMPK. Separately, WDR26 has been linked to autophagy, but with the opposite effect. In H9c2 cells (rat cardiomyoblasts), WDR26 was shown to promote hypoxia-induced autophagy by increasing Parkin translocation at mitochondria and increasing the general ubiquitination of mitochondrial proteins [137].

Several connections of the complex with the WNT pathway have been established. A recent report claimed that RMND5A-MAEA can directly ubiquitinate β-catenin; however, no in vitro ubiquitin assay or binding assay was conducted [138]. The same group previously published that WDR26 associated with Axin, but not with β-catenin [97]. The depletion of WDR26 increased β-catenin stability in *X. laevis* and in WNT-stimulated HEK293 cells independently of GSK3β, and regulated β-catenin ubiquitination if co-expressed with Axin. Interestingly, the entire complex was found in the Axin interactome [139], and MAEA and WDR26 were present in the APC interactome with decreased binding after WNT stimulation [140]. In *D. melanogaster*, β-catenin accumulates in RanBP9 null terminal filament cells of the germ stem cell niche [141].

Some complex members have been associated with the activation of apoptosis in response to cellular stress. In response to IR, RanBP9 has been reported to be phosphorylated in an ATM-dependent manner and initially predominantly nuclear immediately after IR treatment, but then increasingly cytoplasmic as treatment is prolonged [49,142]. At 72 h of IR treatment, RanBP9 is recruited to perinuclear aggresomes [143]. Studies in lung cancer cells showed that RanBP9 is essential for DNA damage response activation, homologous recombination DNA repair, and sensitivity to genotoxic stressors such as IR and cisplatin treatment [49,144]. In *Ranbp9* germ cell knockout testes, enhanced apoptosis of spermatocytes and defective DNA repair is also observed [102]. On the other hand, RanBP9 has been shown to be pro-apoptotic in a variety of cell lines via activation of the intrinsic pathway, as well as through other means, such as regulation of the MAPK pathway, aggresome formation, the activation of cofilin, and interactions with p73 and TSSC3 [132,142,143,145,146,147]. In keratinocytes, ARMC8 expression had a subtle positive effect on apoptosis induction in response to ultraviolet B radiation [148]. Meanwhile, WDR26 expression inhibited oxidative-stress-induced cell death in SH-SY5Y cells and cardiomyocytes [149,150].

Interestingly, the knockdown of RanBP9 enhances IR-induced senescence in a cell-type-dependent manner [49]. This may be linked to the persistence of the DNA damage response activation in those cells. Decreased CTLH-complex-mediated ubiquitination of senescence inducer HBP1 could also contribute to the phenotype in the RanBP9 knockdown cells [16,151]. The effect on senescence and the positive and negative regulations of pathways and processes discussed above suggests that the relationship of CTLH complex in cancer development is context-dependent (reviewed in [152]). A tumour-promoting or -suppressive role for the CTLH complex likely depends on tissue origin, which subunit is altered, the stage of tumorigenesis, and the molecular rewiring of pathways in the context of other mutations in cancer cells.

### 5.5. Functions and Disease Implications in the Central Nervous System

Beyond roles in the development of the brain, a few complex subunits have been linked to neuron signaling and neurodegenerative diseases. For example, in the mouse brain, muskelin is required for normal hippocampal network oscillation and for controlling lysosomal degradation of the cellular prion protein (PrPC) and GABA_A_ receptor (GABA_A_R) [153,154]. Both muskelin and RanBP9 have separately been shown to associate with amyloid precursor protein (APP) [154,155]. In RanBP9-overexpressing mice, APP processing and Aβ generation is elevated, resulting in the increased deposition of amyloid plaques (a hallmark of AD) [110,155,156]. RanBP9 overexpression may also contribute to AD progression by stabilizing Tau protein through interaction with Hsp90/Hsc70 [157]. Although no other complex member has been functionally or genetically linked to AD pathogenesis, UBE2H mRNA is significantly higher in the blood of AD patients [158]. It remains unclear if there are functional relationships between RanBP9, muskelin, and other complex members in the adult brain.

### 5.6. Immune System

There are some reports of roles of CTLH complex members in immunology, although nothing linking the entire complex. The UBE2H promoter contains an NF-κB binding site and is upregulated by the proinflammatory cytokine tumor necrosis factor α (TNF-α) as part of an overall increased ubiquitin conjugating activity observed upon TNF-α treatment [159]. A compelling study discovered that RanBP9 is part of a complex with AXL and LRP-1 that facilitates dendritic cell efferocytosis and antigen cross-presentation to T cells [160]. Additionally, RanBP9 was shown to interact with TRAF6 and suppress the TRAF6 activation of NF-κB signaling [161]. In *D. melanogaster*, the RanBP9/10 homologue was identified as a negative regulator of the cytokine-activated Janus kinase (JAK)/signal transducer and activator of the transcription (STAT) pathway [162].

A connection with viruses was suggested by the presence of the CTLH complex subunits in the interactomes of viral proteins from severe acute respiratory syndrome coronavirus 1 [163,164], Kaposi’s sarcoma-associated herpesvirus [165], and β-herpesvirus human cytomegalovirus [166]. Functionally, RanBP9 and RanBP10 have been identified as host proteins required for viral replication [130,131,167]. Overall, these studies provide evidence that the CTLH complex is involved in immune and viral regulations, although the ubiquitin activity has not yet been implicated.

### 5.7. Endocytosis

Some complex members have been implicated in the internalization of various proteins and endocytosis/lysosomal pathways, an intriguing connection to yeast GID complex regulation of Fbp1 degradation in the vacuole. RanBP9 modulates APP, LRP, and β1-integrin endocytosis in neurons [110,155]. ARMC8 has been shown to promote the interaction of the endosomal sorting complex required for transport (ESCRT) complex with ubiquitinated proteins [168]. As mentioned, muskelin promotes the internalization and degradation of GABA_A_R in mouse neurons [153]. Muskelin interacts with GABA_A_R at the plasma membrane rich in F-actin, where the two proteins associate with Myosin VI. There, muskelin bridges associations of GABA_A_R with dynein and promotes transport in a multivesicular body and subsequent degradation in the lysosome, instead of recycling back to the membrane. It is a process quite reminiscent of the yeast GID-complex-mediated internalization of Fbp1 and subsequent delivery to the vacuole (which, of course, does not involve muskelin) and awaits further investigation to confirm whether other CTLH members are involved in this internalization and transport mechanism.

## 6. Conclusions

In multiple species, it is clear that the CTLH complex is in control of a variety of essential pathways and key biological responses. Its structural and compositional complexity may be related to these diverse functions. A paradigm has emerged of activation of the complex in response to a stimulus, causing the ubiquitination of targets that induces a biological change or cellular adaptation, followed by inactivation of the complex when it is no longer needed (Figure 6). In *S. cerevisiae*, complex activation includes the induction of substrate receptors Gid4 and Gid10 during cellular stress, which is then followed by their own proteasomal degradation [10,39,73]. In *D. melanogaster*, translational upregulation of the UBE2H orthologue, the E2-conjugating enzyme for the complex, during MZT, activates the complex to ubiquitinate its substrates at a precise time [45,96]. Shortly after this event, the presumed substrate receptor for the CTLH complex in this context, muskelin, is rapidly degraded. Interestingly, UBE2H is induced by TNFα treatment and during erythroid differentiation, so regulation of its levels may be a common mechanism for complex activation [159,169]. Furthermore, some subunits have been demonstrated to be regulated by microRNA [170,171,172], subcellular localization [142,173,174], or post-translational modification [48,49], all of which could conceivably act to activate/inactivate the complex or direct it towards specific substrates.

In some cases, the ubiquitin ligase activity of the complex has been associated with regulation, but several processes, such as the implication of the complex in neurodevelopment and neurodegeneration, are missing mechanistic details of the ubiquitination events. Now armed with a much better understanding of the structure and activity of the complex and better resources, we anticipate that much functional insight will be revealed in the near future. One outstanding question is the role of the various protein interaction domains found in several CTLH complex subunits (e.g., SPRY, β-propellers, and discoidin) and whether they can act individually for substrate recruitment or in cooperation with GID4. Evidence is emerging for WDR26 and muskelin to function in substrate recruitment [20,45]. Additionally, the identification of non-yeast GID4 substrates, and by extension, the Pro/N-degron pathway, awaits. As highlighted by Schapira et al., the GID4 binding pocket may be amenable for targeted protein degradation molecules [8], offering an opportunity to develop novel therapeutics. Continued insight into CTLH complex structure, functions, and regulation will be essential to make this a possibility.

## Figures and Tables

**Figure 1 ijms-23-05863-f001:**
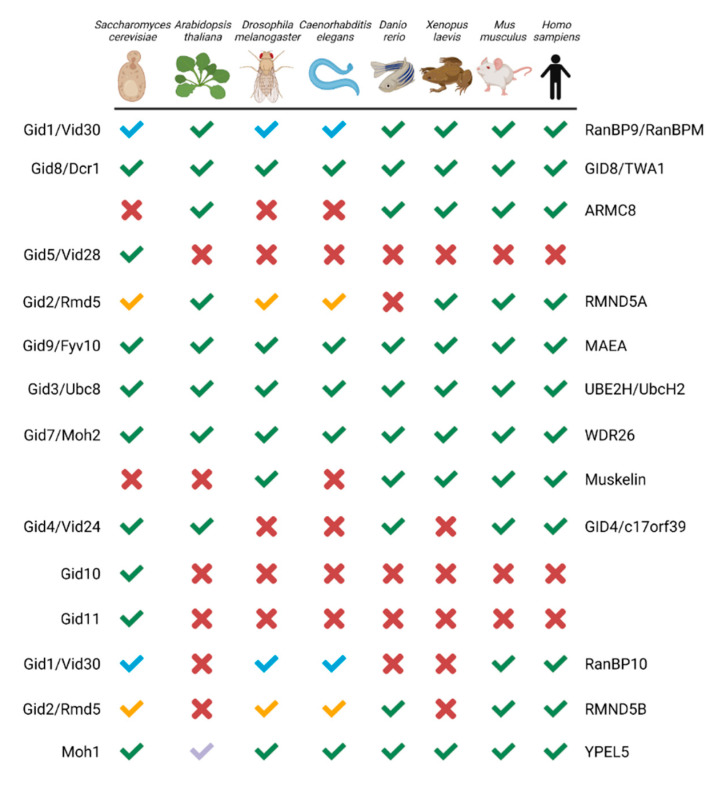
GID/CTLH subunits from yeast to human. Green checkmark or X indicates an orthologue is present or absent, respectively. Paralogues which map to the same gene have a colour-coded checkmark: blue for RanBP9/RanBP10 and yellow for RMND5A/RMND5B. For YPEL5, the light blue checkmark indicates Yipee-like proteins co-purified with RanBPM in *Arabidopsis thaliana*. The yeast and human protein names are indicated on the left and right, respectively. Created with Biorender.com (accessed on 21 April 2022).

**Figure 2 ijms-23-05863-f002:**
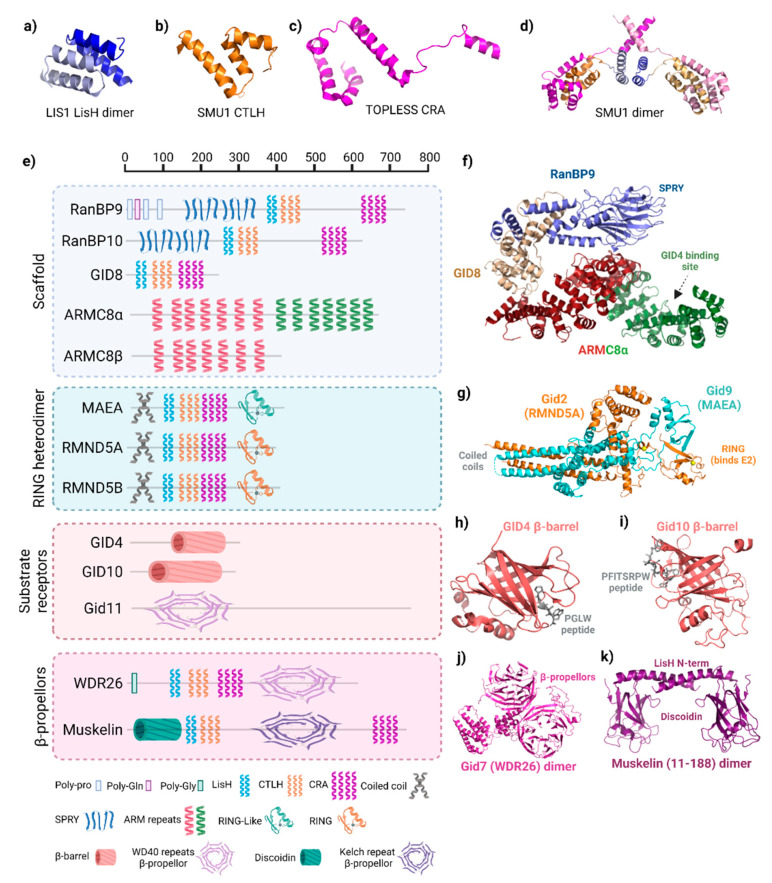
Structure of GID/CTLH subunits. (**a**) Lis1 lissencephaly type-1-like homology (LisH) dimer (PDB: 1UUJ). (**b**) Smu1 C-terminal to LisH (CTLH) motif (PDB: 5EN8). (**c**) TOPLESS CT11-RanBPM (CRA) motif (PDB: 5NQV). (**d**) Structure of SMU1 LisH-CTLH-CRA dimer. LisH (light blue, blue), CTLH (orange, gold), and CRA (violet, pink) in each monomer are shown (PDB: 5EN8). (**e**) Domain organization of GID/CTLH subunits. Scale at the top reflects residue number. All proteins depicted are the human versions, except for Gid7, Gid10, and Gid11, which are *S. cerevisiae.* Legend below denotes the names of each domain and the corresponding symbol, which is representative of the domain structure. Created with Biorender.com (accessed on 21 April 2022). (**f**) Structure of the RanBP9 (blue)–GID8 (gold)-ARMC8α (red/green) scaffold in the human CTLH complex. ARMC8 is split into two colours where red represents structure shared between α and β isoforms, whereas green is only present in α. PDB: 7NSC. (**g**) Structure of the *S. cerevisiae* Gid2 (homologue of RMND5A/B) and Gid9 (homologue of MAEA) RING heterodimer. Zinc ions are coloured yellow (PDB: 7NS4). (**h**) Human GID4 β-barrel structure in complex with a PGLW peptide. (PDB: 6CDC). (**i**) *S. cerevisiae* Gid11 β-barrel structure in complex with a PFITSRPW peptide (7QQY) (**j**) Structure of the *S. cerevisiae* Gid7 (homologue of WDR26) dimer structure (PDB: 7NSB). (**k**) Structure of the N-terminus of a muskelin dimer encompassing the discoidin domain and first helix of the LisH (PDB: 4OYU).

**Figure 3 ijms-23-05863-f003:**
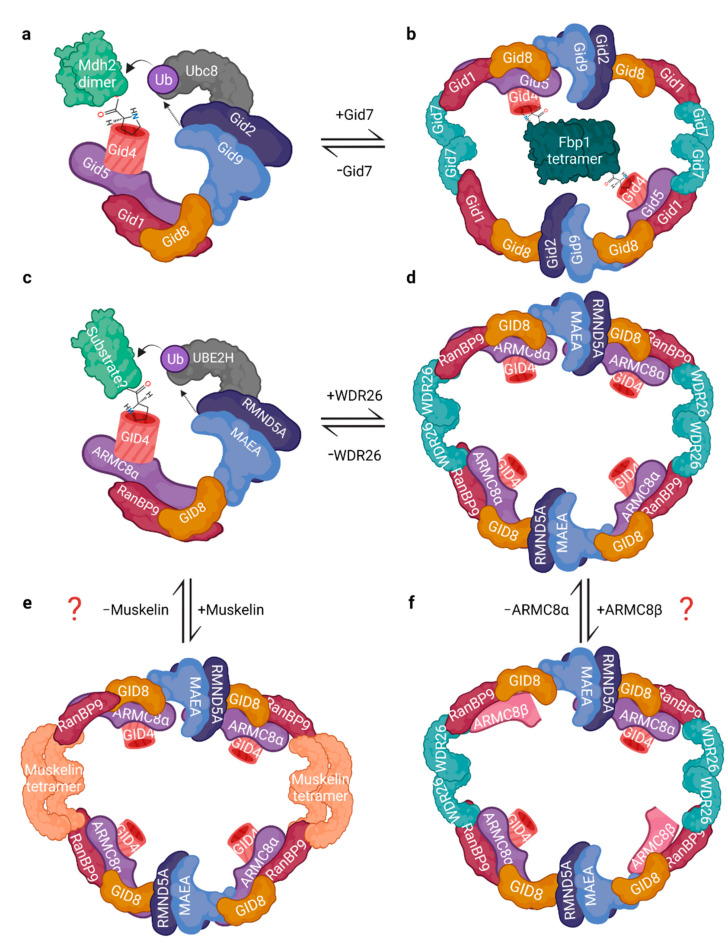
Architecture and assemblies of GID/CTLH complexes. (**a**) Schematic representation of the architecture of the monomeric *S. cerevisiae* Gid complex (without Gid7) binding its cognate E2 Ubc8 and dimeric substrate Mdh2 via N-terminal proline binding the Gid4 β-barrel. (**b**) Supramolecular chelator *S. cerevisiae* GID complex (with Gid7) that encircles its tetrameric substrate Fbp1. (**c**) Architecture of monomeric human CTLH complex (without WDR26 or muskelin) and its cognate E2 UBE2H binding a substrate via GID4. (**d**) Oligomeric assembly of the human CTLH complex containing WDR26. (**e**,**f**) Examples of possible CTLH supramolecular complexes. Additional assemblies may also be formed with RanBP10 and RMND5B paralogues, and different combinations of WDR26, muskelin, and ARMC8 isoforms. For structural details, see text and Qiao et al., 2019, and Sherpa et al., 2021. Created with Biorender.com (accessed on 21 April 2022).

**Figure 4 ijms-23-05863-f004:**
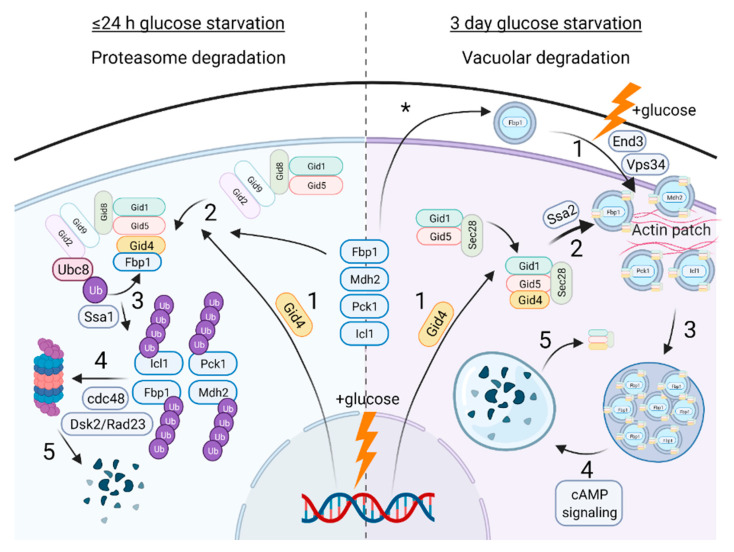
In *S. cerevisiae*, gluconeogenic enzymes undergo distinct glucose-induced, GID complex-dependent degradation mechanisms. Left: 1. In cells starved of glucose for 24 hours, glucose replenishment triggers Gid4 induction. 2. Gid4 associates with the GID complex via Gid5 and recruits gluconeogenic enzymes to the complex. 3. Gluconeogenic enzymes are ubiquitinated by the GID complex via the E2 enzyme Ubc8. This requires the Hsp70 chaperone Ssa1. 4. Polyubiquitinated gluconeogenic enzymes are delivered to the proteasome, which involves cdc48, Dsk2, and Rad23. 5. Gluconeogenic enzymes are degraded. Right: During long-term glucose starvation, gluconeogenic enzymes are secreted as extracellular vesicles in the periplasm (*). 1. Upon glucose replenishment, the gluconeogenic enzymes undergo endocytosis and localize at actin patches, which requires End3 and PI3 kinase Vps34. At the same time, Gid4 expression is induced. 2. At actin patches, 30–50 nm membrane-bound vesicles (named Vid vesicles) are formed with the gluconeogenic enzyme in the lumen and the Gid1–Gid4–Gid5–Sec28 complex on the periphery. Importing the gluconeogenic enzyme substrate into the lumen requires Ssa2. 3. The Vid vesicles aggregate and form endosome-like clusters of varying size that disassociate from actin. 4. The vesicles are delivered to the vacuole, which requires cAMP signaling. 5. The gluconeogenic enzymes are degraded in the vacuole, but other Vid vesicle proteins are returned to the cytosol. In both mechanisms, cells adapt back to a normal state of glycolysis. Created with Biorender.com (accessed on 21 April 2022).

**Figure 5 ijms-23-05863-f005:**
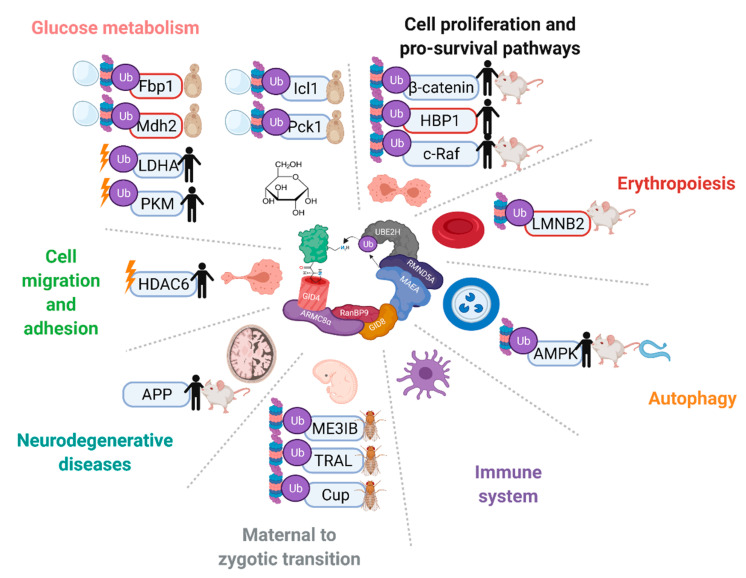
The GID/CTLH complexes are implicated in a variety of pathways and biological processes across multiple species. Proteins that have been reported as targets of the complex are indicated with Ub. In vitro confirmed targets have a red outline. Proteins are marked without a Ub symbol if multiple complex members have been implicated in the regulation, but ubiquitination has not been tested. Proteins are marked with a proteasome and/or vacuole if the complex regulates their degradation, or a lightning bolt if the complex regulates their activity. Species in which regulation on the protein by the CTLH complex has been reported are indicated on the right. Created with Biorender.com (accessed on 21 April 2022).

**Figure 6 ijms-23-05863-f006:**
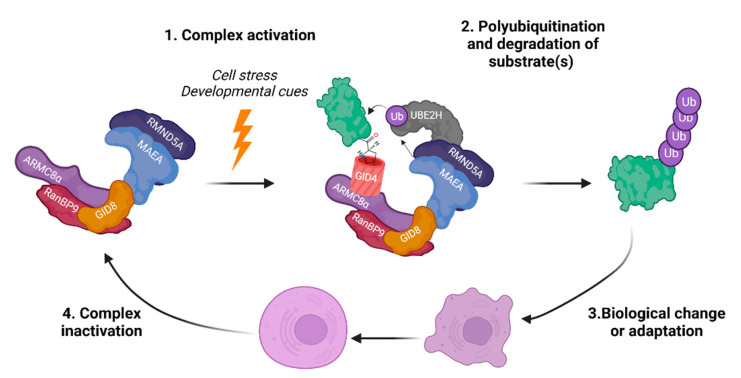
Proposed model of GID/CTLH complex regulation and function in returning to homeostasis. Created with Biorender.com (accessed on 21 April 2022).

**Table 1 ijms-23-05863-t001:** CTLH complex subunit RNA protein-level correlation coefficients across cancer cell lines. Data obtained from Table S4 of Nusinow et al., 2020 [99].

CTLH Complex Subunit	Pearson	Spearman
RANBP9	0.575	0.565
RANBP10	0.480	0.465
WDR26	0.473	0.472
ARMC8	0.408	0.457
GID4	0.344	0.396
RMND5A	0.328	0.295
MKLN1	0.297	0.322
RMND5B	0.159	0.172
GID8	0.105	0.113
MAEA	0.041	0.064

## Data Availability

Not applicable.

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
