# Peer review of "Structural and Functional Insights into GID/CTLH E3 Ligase Complexes"

_ijms, 2022, doi:10.3390/ijms23115863_

Round 1
Reviewer 1 Report
I commend Maitland et al for the well-written essay on GID/CTLH E3 ligase complexes. The review is well balanced and approaches are extremely well described with interesting clinical insights.
Minor point:
In the opinion of this reviewer the manuscript would benefit of a short paragraph in which the role of GID/CTLH subunits in cellular senescence (a stress-induced cell fate which is characterized by resistance to apoptosis) is described.
That is only what I would like to add to this already elegant piece of work.
Author Response
We would like to thank the reviewer for their positive comments on our review manuscript. This reviewer requested ”a short paragraph in which the role of GID/CTLH subunits in cellular senescence (a stress-induced cell fate which is characterized by resistance to apoptosis) is described.”
We have added a short paragraph discussing findings reported in two separate studies that relate to a role for the CTLH complex subunits in cell senescence. This paragraph is located at the end of section 5.6. We feel it is a great addition to this section of the review, and we thank the reviewer for this suggestion.
Reviewer 2 Report
The review by Maitland M. et al. contextualizes relevant and recent research focused on investigate the function(s) and mechanisms of the C-terminal to LisH (CTLH) complex in cellular homeostasis. The manuscript is well written, and the scope of the review is complete and interesting. The figures included are nicely done and meaningful. In general terms, the organization of the review is well performed. This reviewer only find that posttranslational modifications that influence the function and formation of the complex have not been taking into account. In that sense, I find interesting to include the relationship that the most meaningful PTMs have with the complex and the effects that these and other PTMs have in the implcations of the complex in health and disease states as reviewed in the manuscript. I find that with this information this review would be much complete and interesting for the readers of IJMS.
Author Response
We would like to thank the reviewers for their positive comments on our review manuscript.
This reviewer wrote: “This reviewer only find that posttranslational modifications that influence the function and formation of the complex have not been taking into account. In that sense, I find interesting to include the relationship that the most meaningful PTMs have with the complex and the effects that these and other PTMs have in the implcations of the complex in health and disease states as reviewed in the manuscript”.
Response: We agree with the reviewer that this is an important topic that merits to be discussed in the review. We have added a paragraph at the end of section 3.1 that addresses previously reported PTMs on muskelin and RanBP9. However, as the reviewer may know, there is little information on GID/CTLH complex subunits’ PTMs or the stimuli/signaling pathways that promote PTMs on complex subunits, and likewise, very little is known as to how these may modulate complex formation and activity. So the descriptions we added in section 3.1 describe to our knowledge the only examples of specific PTMs and their outcome that have been reported thus far for GID/CTLH complex subunits.
Round 2
Reviewer 2 Report
The changes have been made and the paper can be accepted for publication.